# Temperature dependence of adhesive friction properties of rubber asphalt mortar and its mechanism of action

Yuan Du[1], Ning Li[2]*, Manbin Yang[2], She Shan[1], Hui Dou[2], Yongzhi Chen[1], Xinyuan Liu[1]

1 Gansu Transportation Investment Management Co., Ltd, Lanzhou, Gansu, China, 2 Gansu Industry Technology Center of Transportation Construction Materials Research and Application, Lanzhou Jiaotong University, Lanzhou, Gansu, China

* 273866413@qq.com

## Abstract

The tire-road contact friction system has been widely studied. However, the traditional theoretical model of pavement anti-sliding does not pay attention to the influence of temperature on the tire-road interface friction system. The purpose of this paper is to simulate the adhesive friction process of rubber asphalt mortar under different temperature and different phase conditions by molecular dynamics simulation, so as to explain the adhesive friction mechanism of rubber asphalt mortar. In this study, the molecular model of rubber asphalt mortar with rubber powder content of 0%, 10%, 20% and 30% was first constructed by Materials Studio software. Secondly, the molecular dynamics simulation was carried out by COMPASS II force field, and the glass transition temperature, bulk modulus and Young's modulus of the molecular model of rubber asphalt mortar were calculated. Then, the variation characteristics of the friction coefficient index of the rubber-asphalt mortar-aggregate three-layer friction pair model with temperature were simulated, and the simulation results of elastic modulus and adhesion work were comprehensively analyzed. The results showed that the glass transition temperature of the molecular model of rubber asphalt mortar is between -20°C and 0°C, and the bulk modulus and Young's modulus of rubber asphalt mortar show an upward trend with the increase of rubber powder content and temperature. The friction coefficient of the rubber-asphalt mortar-aggregate three-layer friction pair model increases with the increase of temperature and rubber powder content. The results of Young's modulus and adhesion work simulation show that as the temperature increases, the phase state of the rubber asphalt mortar changes, and the adhesion increases, resulting in an increase in the friction coefficient of the rubber asphalt mortar. In addition, the simulated and predicted values of the friction coefficient of the rubber-asphalt mortar-aggregate three-layer friction pair model with temperature change are well fitted, and the maximum difference is 0.32.

**Data availability statement:** All relevant data are within the manuscript and its Supporting Information files.

**Funding:** Innovation Group in Fundamental Research (25JRRA145), Special Funds for Guiding Local Scientific and Technological Development by The Central Government (22ZY1QA005), Gansu Science and Technology Major Project (22ZD6GA010), Gansu Provincial Key R&D Program(22YF7GA135), the Key Research and Development Program of Ningxia Autonomous Region of China (2022BDE02010), and Lanzhou Science and Technology Project (2022-2-72). The funders had no role in study design, data collection and analysis, decision to publish, or preparation of the manuscript.

**Competing interests:** The authors have declared that no competing interests exist.

## 1. Introduction

Pavement function design should not only meet the requirements of structural durability and driving comfort, but also put forward higher requirements for pavement safety design [1,2]. Anti-skid performance is one of the most important indicators reflecting the safety performance of the road surface. Excellent anti-skid performance of the road surface provides good adhesion between the tire and the road for the vehicle during driving, ensures the friction required for the brake between the tire and the road, and reduces the probability of traffic accidents [3,4]. The friction coefficient between tire and pavement is the main index to measure the skid resistance of pavement, and the tire-road friction can be mainly attributed to the combined effect of adhesion component and viscous component [5–7]. In the field of road engineering, in order to scientifically characterize the adhesion component in the tire-road contact friction system during relatively low-speed sliding, the friction adhesion model between tire rubber and road surface roughness is established by numerical simulation method and mechanical analysis method, and the semi-empirical friction adhesion coefficient is obtained [8–10].

In road engineering, scholars have done a lot of research on the contact friction characteristics of asphalt pavement. Aliha et al. [11] measured the interlayer contact force of the pavement, and found that the interlayer contact state is proportional to the friction coefficient, which effectively improves the pavement stiffness, reduces the deflection value and the vertical compressive strain at the top of the subgrade. Guo et al. [12] studied the directional tire-road contact mechanism to correctly calculate the directional friction coefficient of the road surface. The calculated directional friction coefficient was verified by experimental measurements. Zhang et al. [13] established a friction model considering road surface morphology, tire characteristics, sliding speed and contact pressure. The simulation results are consistent with the experimental results of wet road braking distance. Anahita et al. [14,15] used the data of surface roughness power spectrum and viscoelastic modulus, combined with the improved Persson contact theory to model the real contact area of rubber samples on two surfaces. The friction coefficient of the Vehicle [16,17] experiment was compared with the friction coefficient calculated by the Heinrich/Klüppel model. The relative percentage error between the test and the friction model results was found to be less than 10% on the smooth concrete pavement and less than 20% on the rough pavement. Hamdi and Hossain et al. [18,19] studied the stick-slip problem in the sliding process of rubber tire materials, and found that increasing the thickness of soft matrix inhibited the formation of cracks in the adhesion process, and increased plastic deformation and friction.

The skid resistance of pavement is affected by the characteristics of tire rubber, pavement morphology and the slip velocity of the interface between the two, which is also closely related to the complexity of the tire-road contact problem itself. It is difficult to fully characterize the friction behavior between tire and asphalt pavement from a macro perspective. More and more scholars use molecular dynamics to study the contact friction characteristics of asphalt pavement. In order to study the friction



characteristics of tire tread rubber, Zhao et al. [20] carried out various contact conditions, and then proposed a unified friction model to describe the nonlinear relationship between rubber friction and contact pressure and sliding speed. In order to clarify the relationship between the gene of tire pavement material and its friction performance, molecular dynamics simulation was used to simulate the constrained shear simulation to simulate the friction process between tire rubber and aggregate. Raj Chawla et al. [21] studied the tribological behavior of carbon nanotubes reinforced styrene-butadiene rubber by molecular dynamics simulation. It was found that the friction coefficient and wear rate decreased with the increase of sliding speed. Yu et al. [22] found that the friction behavior between tire and road surface with surface texture will be more complex, and the existing models are still lacking in characterizing the friction mechanism of tire-road. Based on this, Sun et al. [23] used molecular dynamics simulation analysis method to establish three-dimensional monomer model and interface contact model of tire and aggregate, and studied the microstructure and contact characteristics of tire and aggregate at nano scale.

In summary, it is found that scholars often establish a model of adhesion friction coefficient based on the micro-roughness, real contact area, interface shear pressure, speed and other parameters of the mixture surface when establishing the tire-road friction adhesion model. The above model does not pay attention to the influence of temperature on the tire-road interface friction system, and also fails to reflect the influence of asphalt mortar between aggregate and tire on the tire-road contact friction model [24–28]. At the same time, because rubber is not only a highly elastic polymer material with significant phase change characteristics [29,30], but also a homogeneous material with tire rubber, the adhesion and friction properties of rubber asphalt mortar and rubber tire are studied in order to obtain more significant test rules. Therefore, it is necessary to simulate the adhesive friction process of rubber asphalt mortar under different temperature and different phase conditions by molecular dynamics simulation, and to explain the adhesive friction mechanism of rubber asphalt mortar.

In this study, Materials Studio 2020 software was used to construct the molecular model system of rubber asphalt mortar and rubber asphalt mortar-aggregate interface under the condition of real powder-binder ratio, and the model was verified and the physical properties were simulated. On this basis, a three-layer friction pair model of rubber-asphalt mortar-aggregate was established to simulate the adhesive friction performance of the interface friction system. The variation law of the friction performance of rubber asphalt mortar under different temperature conditions and different rubber powder content conditions was analyzed, and the mechanism of the adhesive friction performance of rubber asphalt mortar was explained.

## 2. Materials and experiments

### 2.1. Raw materials

(1) Asphalt

In this study, the neat asphalt is Zhenhai 90# asphalt provided by Gansu Highway Maintenance Technology Research Institute Co., Ltd., and its technical indicators refer to the 'Highway Engineering Asphalt and Asphalt Mixture Test Procedure' (JTG E20-2011) [31]. The test results are shown in Table 1.

(2) Mineral powder

In this study, limestone powder was selected. The powder is dry and clean. According to the relevant provisions of 'Highway Engineering Aggregate Test Procedures' (JTGE42-2005) [32], the performance test is carried out. All indicators meet the requirements of the specification, and the relevant test results are shown in Table 2.

(3) Rubber powder

The rubber powder used in the study was from 40 mesh waste oblique tire rubber powder produced by normal temperature crushing process of Gansu Wuwei Xinhaoyuan Environmental Protection Technology Co., Ltd. The test results are

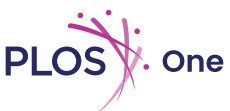

**Table 1. Properties of asphalt.**

| Properties | | Value |
|---|---|---|
| Penetration (25°C, 100 g, 5s) (0.1 mm) | | 87.6 |
| Softening point (°C) | | 46.8 |
| Ductility (5 cm/min, 15°C) (cm) | | >100 |
| Density (g/cm³) | | 1.039 |
| Brookfield viscosity (135°C) (mPa·s) | | ≤3 |
| After RTFOT(163°C,85min) | Quality change (%) | ±1 |
| | Residual penetration ratio (25°C) (%) | ≥65 |
| | Residual ductility (25°C) (cm) | ≥8.0 |

**Table 2. Properties of mineral powder.**

| Properties | | Value |
|---|---|---|
| Apparent density (g/cm³) | | 2.725 |
| Water content (%) | | 0.42 |
| Hydrophilic coefficient | | 0.60 |
| Particle size range (%) | <0.6 mm | 100 |
| | <0.15 mm | 98 |
| | <0.075 mm | 93.2 |

shown in Table 3, and all the technical indicators meet the requirements of 'Rubber Asphalt Pavement Technical Standard' (CJJT273-2019) [33].

## 2.2. Friction coefficient of rubber asphalt mortar

In this study, the HP-MXD-01C friction coefficient tester was used to test the friction coefficient curve and the corresponding static friction coefficient and dynamic friction coefficient when the flaky sample slides on the surface of other materials. The instrument meets the requirements of GB/T1006-1988 'Determination of plastic-film and sheet-friction coefficient' [34] (see Table 4).

During the test, the surface of the rubber asphalt mortar friction plate is first upward and fixed flatly on the horizontal test bench, so that the sample is parallel to the length direction of the test bench. In order to simulate the contact process between the rubber asphalt mortar on the surface of the mixture and the tire under real working conditions, the synthetic rubber block with similar composition to the commonly used car tire was selected as the friction pair material with relative displacement to the sample. After that, the friction pair material is fixed on the sliding rod, and the sample is parallel to the direction of the sliding rod motion, and the force measuring system is not stressed. Click the 'Start' button to start the experiment. When the number of tests reaches the set number of times, the test is automatically completed. To perform the next set of tests, press the 'Reset' button to return to standby.

## 2.3. Model establishment

### 2.3.1. Molecular model for asphalt.
Asphalt is a complex chemical mixture containing millions of macromolecular hydrocarbons and non-metallic derivative molecules containing sulfur, nitrogen and oxygen atoms. Therefore, it is difficult to accurately characterize all molecular compositions and accurately evaluate their properties. In this study, according to the Corbett separation method [35], the asphalt is divided into four components of SARA, such as saturates, aromatics, resins and asphaltenes, which provides an effective method for studying the typical structure of asphalt. In



**Table 3. Properties of rubber powder.**

| Properties | Value |
|---|---|
| Relative density (g/cm³) | 1.12 |
| Moisture content (%) | 0.55 |
| Iron content (%) | 0.01 |
| Fiber content (%) | 0.07 |
| Ash (%) | 7.28 |
| Acetone extract (%) | 7.26 |
| Carbon black content (%) | 31 |
| Rubber hydrocarbon content (%) | 55 |

**Table 4. Performance index of friction coefficient meter.**

| Object name | Weighter weight | Slider moving speed | Test force accuracy | Maximum specimen size | |
|---|---|---|---|---|---|
| | | | | Upper sample | Lower sample |
| Technical specification | 200±2 g | 100±10 mm/min | ±0.02 N | 63*63 mm | 80*200 mm |

order to understand the physical, mechanical and rheological properties of asphalt, a 12-component molecular model of SHRPAAA-1 asphalt based on SARA component developed by Li and Greenfield [36] is used in this paper. The molecular characteristics of asphalt components are shown in Table 5, and their molecular structures are shown in Fig 1.

**2.3.2. Molecular model for mineral powder.** Because the main component of mineral powder is $SiO_2$, this study chooses $SiO_2$ molecule to simplify the molecular structure model of mineral powder to represent mineral powder (see Fig 2) [37]. Firstly, the unit crystal silicon was selected from the structural database of Materials Studio, and its lattice parameters were a=b=4.909A, c=5.402A, c=β=90°, y=120°. Then a-silica supercell model was constructed with supercell parameters of A=2, B=1 and C=2. Finally, a silica nanocluster molecule with a radius of 5A was established, and the atoms outside the radius were discarded 301. In addition, -H and-OH are added to the surface of oxygen atoms and silicon atoms in the silica cluster structure, respectively, so that the silica nanoclusters are electrically neutral as a whole. The silica nanoclusters are composed of 16 Si atoms, 50 O atoms and 24 H atoms.

**Table 5. Molecular structure of asphalt components.**

| Component | Molecular quantitative | Molecular formula | Molecular weight (g/mol) | Mass fraction (%) |
|---|---|---|---|---|
| Saturate | 4 | $C_{30}H_{62}$ | 422.9 | 5.2 |
| | 4 | $C_{35}H_{62}$ | 482.8 | 5.8 |
| Aromatic | 11 | $C_{35}H_{44}$ | 464.8 | 15.7 |
| | 13 | $C_{30}H_{46}$ | 406.8 | 16.2 |
| Resin | 4 | $C_{36}H_{57}N$ | 530.9 | 6.2 |
| | 4 | $C_{40}H_{60}S$ | 572.9 | 7.0 |
| | 5 | $C_{29}H_{50}O$ | 414.7 | 6.4 |
| | 4 | $C_{40}H_{59}N$ | 554.0 | 6.8 |
| | 15 | $C_{18}H_{10}S_2$ | 290.4 | 13.4 |
| Asphaltene | 3 | $C_{42}H_{54}O$ | 575.0 | 5.3 |
| | 2 | $C_{66}H_{81}N$ | 888.5 | 5.5 |
| | 3 | $C_{51}H_{62}S$ | 707.2 | 6.5 |

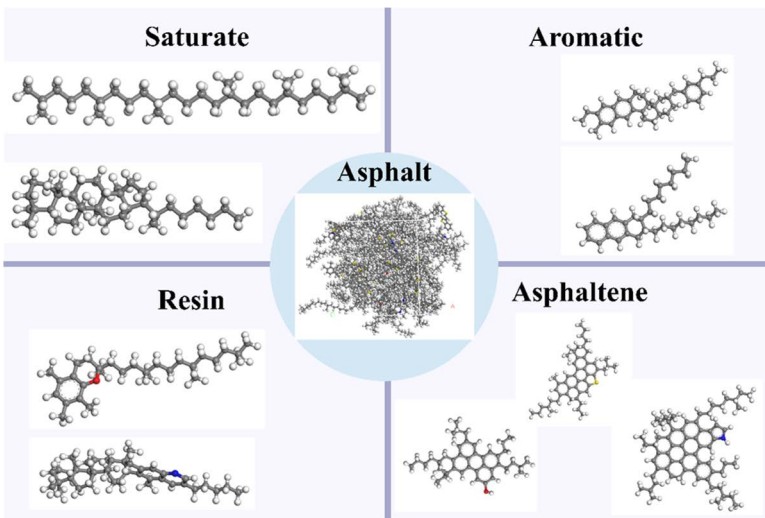

**Fig 1. Asphalt components molecular model.**

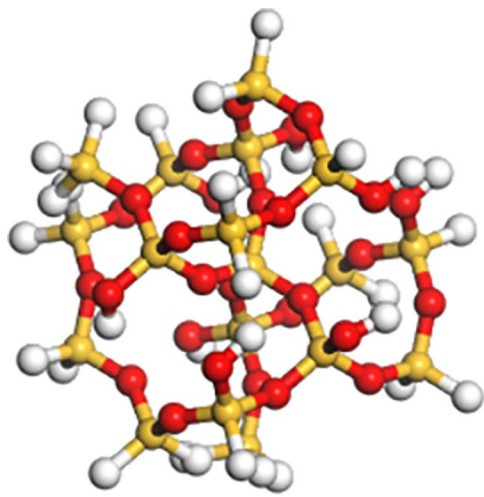

**Fig 2. Mineral powder molecular model.**

**2.3.3. Molecular model for rubber.** In this study, waste rubber powder was selected as the simulation object, which was mainly composed of natural rubber (NR) and synthetic rubber (SBR). The proportion of natural rubber and synthetic rubber was 20: 80. The MS polymer modeling tool was used to construct the two polymers respectively, and then the software amorphous blend modeling function was used to assemble according to the actual proportion of the components.

Firstly, two rubber representative molecular models were selected. One is natural rubber. The macromolecular chain structure unit of natural rubber is isoprene, which is mainly composed of cis-1,4-polyisoprene. In the process of molecular simulation, the selection of molecular polymerization is very important. In this paper, the single chain model of natural rubber is defined as 20, and the molecular model of natural rubber NR monomer is constructed by using the Visualizer module in MS. The second is styrene-butadiene rubber. The macromolecular chains of styrene-butadiene rubber are

mainly butadiene and styrene, which belong to the amorphous polymer of butadiene and styrene. The molecular shape is mostly linear and the structure is complex. Among them, butadiene also includes trans-1,4 structure, cis-1,4 structure and 1,2-polybutadiene structure.

Using the Build Polymer function in Material Studio software, the different components of the whole molecular chain can be randomly and freely connected to form monomer molecules, and the SBR monomer molecular model of butadiene and styrene random copolymer can be constructed.

Then, the two monomer molecular models were assembled into amorphous rubber molecules with three-dimensional periodic boundary conditions at a ratio of 20: 80. Finally, under the regular (NVT) ensemble, the COMPASS force field was selected, and simulated annealing was performed in the temperature range of 300~500K for 5 times to optimize the system to a reasonable structure, as shown in Fig 3.

**2.3.4. Molecular model for rubber asphalt mortar.** In the construction process of molecular model for rubber asphalt mortar, asphalt molecules, rubber molecules and silica molecules are randomly put into a cubic empty box with an initial density of $0.6\,g/cm^3$ to ensure that each molecular model is evenly distributed in the model and the molecular chain was not distorted. The forming process of rubber asphalt mortar is shown in Fig 4. Under the condition of powder-binder ratio of 1.0, the asphalt mortar model under the condition of real powder-binder ratio is established.

After the model was constructed, the Smart algorithm was used to geometrically optimize the model and iterate for 5000 times to minimize the system energy and optimize the model structure of the asphalt mortar. Secondly, the annealing treatment was carried out, and the temperature range was set from 273.15K to 1800.15K. The heating and cooling were repeated for five times to remove the unstable configuration. Then, at 298.15K, a standard atmospheric pressure of

**Fig 3. Rubber molecular model.**

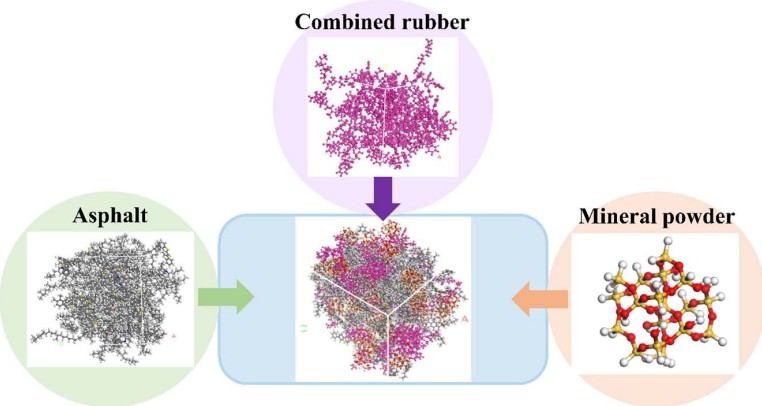

**Fig 4. Molecular model of the rubber asphalt mortar.**

500 ps NPT ensemble dynamic equilibrium was carried out to obtain the ideal molecular structure. Finally, a 500 ps NVT ensemble was performed at 298.15 K to obtain a final equilibrium structure with stable volume and energy. The asphalt mortar model with different rubber powder content is shown in Fig 5.

## 3. Methodology

### 3.1. MD simulated methods

In molecular dynamics simulation, the evolution of the system with time is calculated by solving the Newton's motion equation of classical mechanics to calculate the force of each atom in the system at different times and the related motion trajectory at the corresponding time [38]. In a system of N atoms, each atom $i$ follows Newton's second law, as shown in Eq. (1).

$$F_i(t) = m_i a_i = m_i \frac{d^2 r_i(t)}{dt^2}$$

(1)

where $F_i(t)$ is the force of atom $i$ at time $t$, $m_i$ is the mass of atom $i$, $a_i(t)$ is the acceleration of atom $i$ at time $t$, $r_i(t)$ is the vector displacement of atom $i$ at time $t$. It is not difficult to find that molecular dynamics is a deterministic algorithm, which solves the spatial distribution of the system phase by motion integration, and then obtains the change process of the system with time.

(1) Field of force

The force field is the basis for describing and calculating the interaction between molecules and atoms in the system. The evaluation and modeling of macroscopic properties of materials depend on the force field and molecular structure.

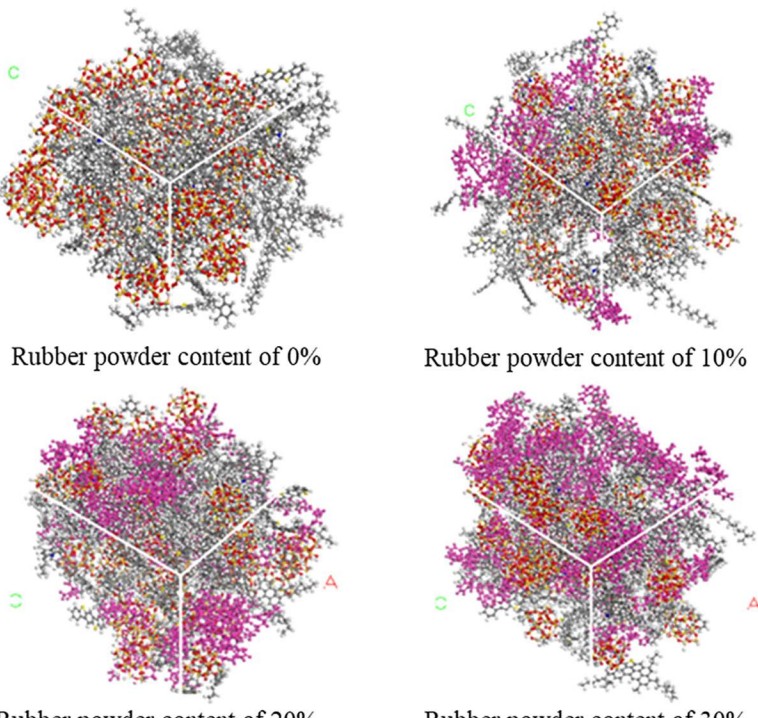

Rubber powder content of 0%   Rubber powder content of 10%

Rubber powder content of 20%   Rubber powder content of 30%

**Fig 5. Molecular model of rubber asphalt mortar with different rubber powder content.** (a) Rubber powder content of 0% (b) Rubber powder content of 10% (c) Rubber powder content of 20% (d) Rubber powder content of 30%.

Therefore, selecting the appropriate force field for characterizing the material properties is the key to molecular dynamics modeling. As the force field becomes more and more extensive and mature, the calculation accuracy and efficiency of molecular dynamics modeling have been steadily improved.

(2) Periodic boundary condition

The periodic limit condition is a structural model composed of the same periodic grid [39]. After the introduction of periodic boundary conditions, the simulation system becomes a part of the infinite molecular system with the same properties, referred to as the central cell. In the dynamic modeling process, microscopic particles can not only move within the central cell, but also be released from the central cell to the adjacent cell. At the same time, adjacent cells can also enter the central cell, which can be called the nearest mirror method.

(3) Ensemble

Ensemble refers to a large number of independent systems with identical properties and structures, in various dynamic states and subject to certain macro constraints under certain macro constraints. It is usually used to describe the statistical laws of thermodynamic systems. The system is only a means of expressing statistical theory, not an objective physical object. This paper focuses on the adhesion and failure behavior of asphalt mortar-aggregate interface. The NVT system can adjust the temperature of the system by hot bath to simulate the change of interface behavior under different temperature conditions. Therefore, this paper chooses NVT system for dynamic simulation.

(4) Energy minimization

Before performing molecular dynamics simulation calculations, it is usually necessary to optimize and adjust the system structure model construction to obtain a stable and reasonable configuration, and eliminate the molecular overlapping configuration during the system construction process to ensure subsequent simulation calculations. In this paper, the modules of the Materials Studio software Forcete were used for geometric optimization, and the Smart algorithm was used for structural optimization to minimize the energy of the asphalt mortar interface model.

### 3.2. Evaluation indicators

(1) Radial distribution function

In this study, the density of asphalt and the radial distribution function g(r) within the molecule were selected to verify the accuracy of the molecular model. The radial distribution function reflects the distribution and aggregation of molecules inside the material. The radial distribution function g(r) is shown as Eq. (2).

$$g(r) = \frac{dN}{\rho 4\pi r^2 dr}$$

(2)

where $\rho$ is the density of the system and $N$ is the number of particles.

(2) Glass transition temperature ($T_g$)

Glass transition temperature ($T_g$) is suitable for asphalt research. As a typical amorphous polymer mixture, asphalt is usually in the glass state, high elastic state, viscoelastic state, viscous flow state and fluid state. The glass transition temperature ($T_g$) is the temperature at which the polymer changes from a high elastic state to a glassy state. At the molecular level, the glass transition temperature ($T_g$) is the temperature at which the polymer chain and segment motion state change.

(3) Bulk modulus

Bulk modulus is a stiffness measurement of the ability of a material to withstand compressive or tensile volume changes in three directions. The calculation formula of bulk modulus is shown in Eq. (3).

$$K = \lambda + \frac{2}{3\mu}$$

(3)

where $\lambda$ and $\mu$ are Lame constants.

(4) Young's modulus

Young's modulus, also known as elastic modulus, is the stiffness measurement of elastic materials. It is the ratio of stress to corresponding strain in one direction. The calculation formula of Young's modulus is shown in Eq. (4).

$$E = \frac{\mu(3\lambda + 2\mu)}{\lambda + \mu}$$

(4)

where $\lambda$ and $\mu$ are Lame constants.

## 4. Results and discussion

### 4.1. Model reliability verification

**4.1.1. Molecular order verification.**  The asphalt density after relaxation of NVT ensemble is 1.030 g/cm³, which is close to the asphalt density 1.032 g/cm³ used in the test, and the asphalt density obtained is basically in line with the actual situation. The interaction force between the molecules of the material can generally be reflected by g(r), and the size of g(r) indicates the degree of disorder of the structure. The Forcite calculation module is used to extract g(r), and the results are shown in Fig 6. For asphalt, the hydrogen bond between molecules can affect the range of 2.6-3.1Å, and the van der Waals force can affect the range of 3.1-5Å. It can be seen from the diagram that the g(r) of each component of asphalt begins to decrease at about 3 Å, and the curve tends to be gentle and close to 1 after 5 Å, which is consistent with the actual situation.

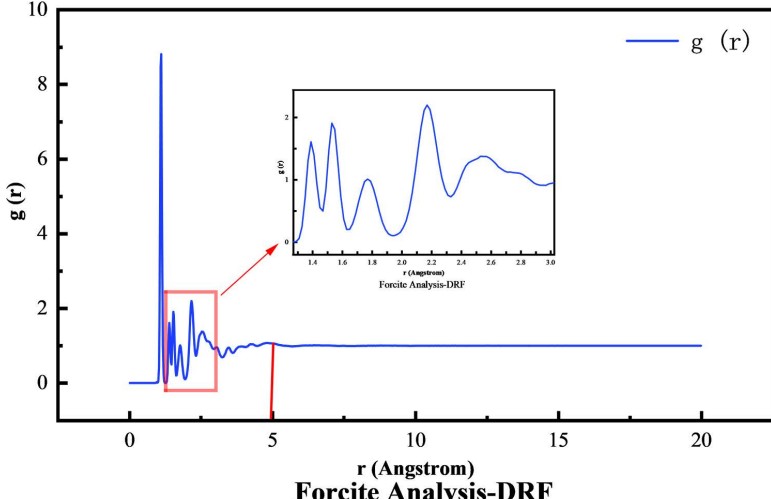

**Fig 6. Internal radial distribution function diagram of asphalt molecular model.**

Therefore, by comparing the simulation results of asphalt density and radial distribution function, it is considered that the model can reflect the real asphalt molecules. The raw data for Fig 6 are available in S1 Fig.

**4.1.2. Glass transfer temperature.** Glass transfer temperature (Tg) is an important standard to evaluate the low temperature performance of pavement. When the temperature drops below Tg, the asphalt material will completely become a brittle material, almost losing the deformation ability, and it is prone to brittle failure under conventional load. Therefore, it is very important to clarify the change of glass transfer temperature of rubber asphalt mortar. Therefore, the geometric optimization and annealing treatment of the molecular model of rubber asphalt mortar are carried out to make the molecular model reach a stable state, and then the molecular dynamics simulation of the model at a temperature of -60°C~200°C (gradient of 20°C) is carried out respectively. The obtained trajectory file is analyzed, and the variation law of the density of the molecular model of rubber asphalt mortar with temperature is drawn, as shown in Fig 7. The raw data for Fig 7 are available in S2 Fig.

It can be seen from Fig 8 that the density of rubber asphalt mortar with rubber powder parameter of 0% is the largest, and the higher the rubber powder content, the smaller the density of rubber asphalt mortar. With the increase of temperature, the density of three kinds of rubber asphalt mortar decreased. This is because the increase of temperature makes the kinetic energy of each molecule in the rubber asphalt mortar model increase, and the molecular motion is more active, which makes the volume expand and the density decrease. In the range of -60°C~-20 °C, the density decreases slowly with the increase of temperature. In the range of -20°C~0°C, obvious changes have taken place. This may be because when the temperature is low, the intermolecular force is strong, and the temperature is slightly increased, which is not enough to make the molecule break away from the range of interaction force. When the temperature reaches a certain value, the kinetic energy converted by the molecular absorption heat energy is greater than the interaction energy between the molecules, the molecules are separated from the interaction force range, and the density is greatly reduced. In the range of -20°C~0°C, obvious changes have taken place, and the glass transfer temperature of the three rubber asphalt mortar molecular models is in the range of -20°C~0°C.

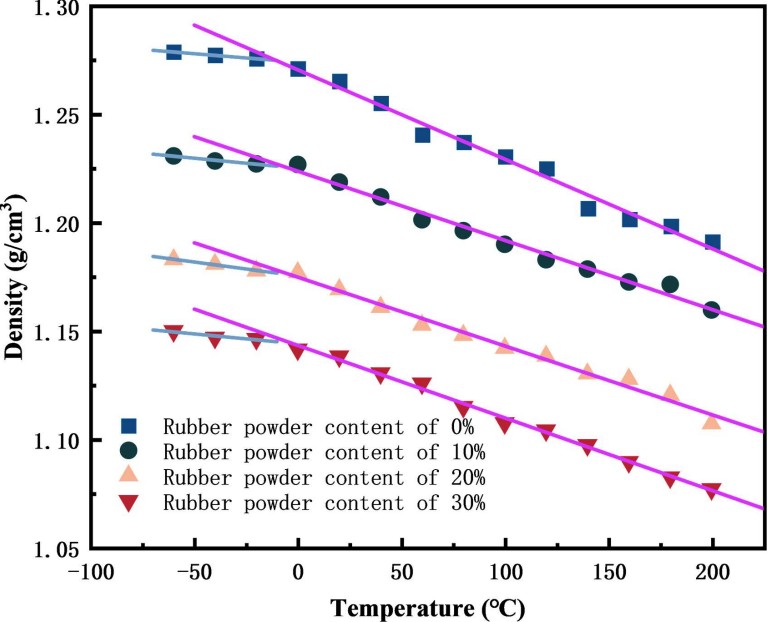

**Fig 7. The density of rubber asphalt mortar molecular model changes with temperature.**

**4.1.3. Bulk modulus and Young's modulus.** The Mechanical Properties analysis in Forcite was carried out on the four rubber asphalt mortar molecular models with rubber powder content of 0%, 10%, 20% and 30%. The Lame constants $\lambda$ and $\mu$ of different rubber asphalt mortars at different temperatures were obtained by controlling the simulation temperatures at -60°C, -40°C, -20°C, 0°C, 20°C, 40°C and 60°C, respectively. The calculation equations for bulk modulus $K$ and Young's modulus $E$ of rubber asphalt mortar are shown in Eqs (3) and (4).

It can be seen from Figs 8 and 9 that when the temperature is the same, the higher the rubber powder content, the smaller the Young's modulus and the smaller the bulk modulus. This may be the reason why the density of rubber asphalt mortar with 0% rubber powder content is the largest at the same temperature. With the increase of temperature, the bulk modulus and Young's modulus of rubber asphalt mortar show a decreasing trend. This is because under low temperature conditions, the isothermal compression coefficient of asphalt molecular model is low, so the modulus of rubber asphalt mortar is high and not easy to be compressed. With the increase of temperature, the isothermal compression coefficient of asphalt molecular model also increases gradually, so that its bulk modulus and Young's modulus show a decreasing trend. In the range of -20°C~0°C, it can be seen that the bulk modulus and Young's modulus of the molecular model of rubber asphalt mortar have a significant change, which may be caused by the glass transfer temperature of rubber asphalt mortar. The raw data for Figs 8 and 9 are available in S3 and S4 Figs.

## 4.2. Rubber asphalt mortar friction system

**4.2.1. Establishment of friction system.** In order to study the effect of rubber powder content on the friction performance of rubber asphalt mortar under actual service conditions, a three-layer friction pair molecular model of rubber/rubber asphalt mortar/aggregate with four different rubber powder contents of 0%, 10%, 20% and 30% is established respectively. The global energy minimum configuration of the rubber asphalt mortar is used as the intermediate layer to form a three-layer friction pair model with the top rubber interface and the $SiO_2$ atomic layer at the bottom. The rubber at the top and the aggregate layer at the bottom are $5.0 \times 5.0 \times 2.0\,nm^3$ and $4.4 \times 4.4 \times 1.6\,nm^3$, respectively. The friction system of rubber asphalt mortar with different rubber powder content is shown in Fig 10.

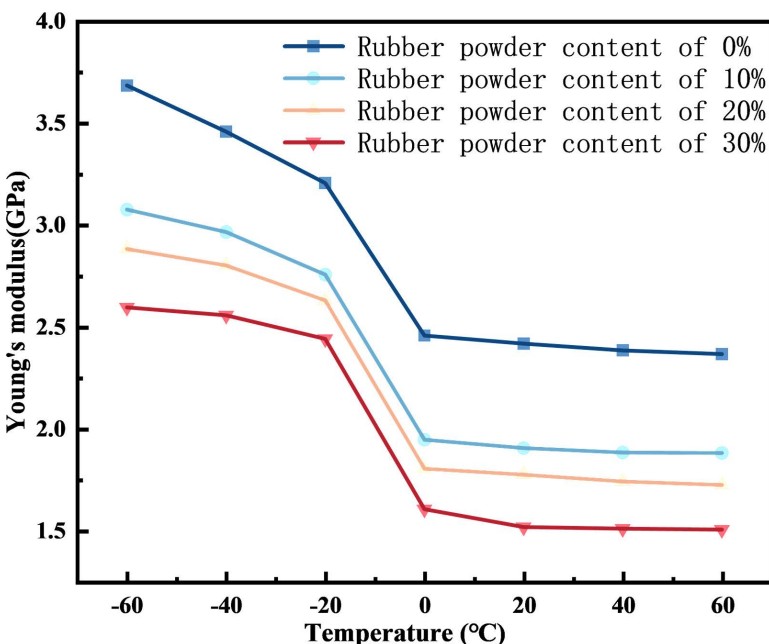

**Fig 8. The change of Young's modulus of rubber asphalt mortar with temperature.**

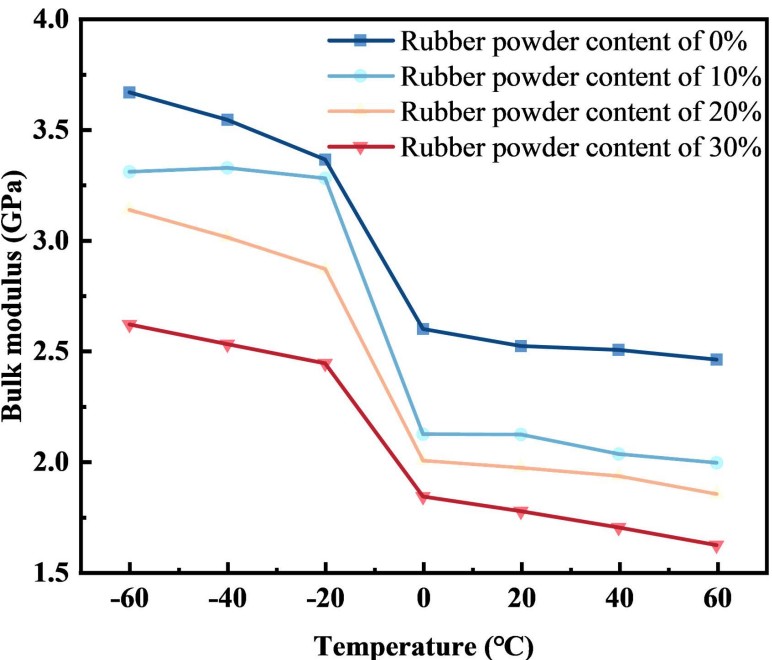

**Fig 9. The change of Bulk modulus of rubber asphalt mortar with temperature.**

**4.2.2. System optimization.** After the model is established, it is necessary to optimize the structure and balance the molecular dynamics of the above friction pair model. Firstly, the Cartesian positions of the upper and lower rubber and aggregate layers are fixed, and the Smart method is used to geometrically optimize the three-layer friction pair model until the energy difference between the two adjacent iteration steps is less than the energy convergence accuracy of 10–5 kal/mol. Subsequently, the friction system is subjected to five annealing cycles (Anneal) to obtain the global optimal configuration.

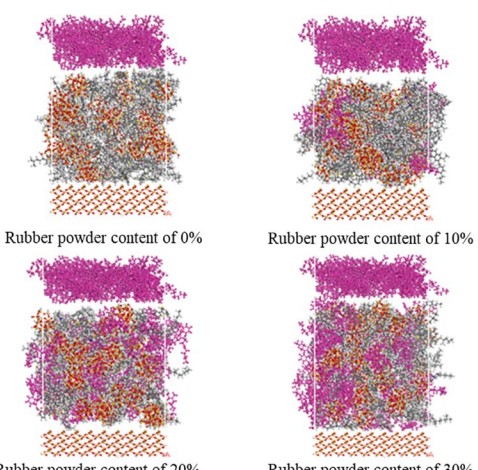

**Fig 10. Schematic diagram of rubber asphalt mortar friction system.** (a) Rubber powder content of 0% (b) Rubber powder content of 10% (c) Rubber powder content of 20% (d) Rubber powder content of 30%.

The annealing cycle is as follows: the initial temperature is set to 300 K, the temperature growth step is 50 K, and the target temperature is set to 1800 K. When the temperature reaches the target temperature, 1 fs is used as a time step, and the temperature is reduced by 50 K each time until the temperature is reduced to 300 K. During the temperature drop, the molecular dynamics equilibrium of the NVT ensemble is performed on the system at each temperature, and a total of 200 ps of molecular dynamics equilibrium is performed in one cycle. After each annealing cycle, the smart method is used to optimize the structure of the system for 5000 steps. The system is continuously cycled for 5 times according to this law. During the annealing process, the molecular system undergoes continuous heating and cooling cycles, the system structure obtains sufficient relaxation, and the system structure tends to be stable.

The lowest energy conformation after annealing is selected, and the Cartesian position of the top rubber and the bottom aggregate is removed. The Confined Shear task in the Forcite Plus module is selected. The initial velocity of the relative movement of the upper and lower metal atomic surfaces is set to be 0.1 Å/s, and the direction extends to the positive and negative directions of the X axis. Then the system is simulated and run for 200 ps under the NVT ensemble at a temperature of 273.15 K. Finally, the simulation results of restricted shear are obtained. The simulation process under different rubber powder content is shown in Fig 11.

### 4.3. Friction behavior analysis of rubber asphalt mortar

**4.3.1. Restricted shear simulation calculation.** According to the friction law proposed by Eder et al. (Eqs (4)–(6)), the friction coefficient of the interface between rubber asphalt mortar and rubber was calculated.

$$\mu = \frac{F_f}{F_n}$$

(5)

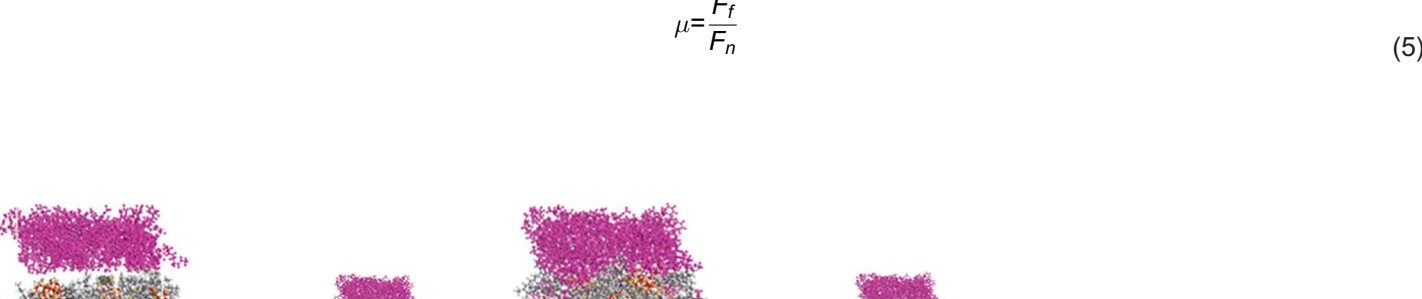

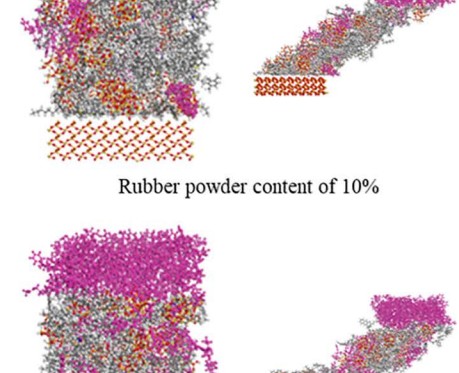

Rubber powder content of 0%

Rubber powder content of 10%

Rubber powder content of 20%

Rubber powder content of 30%

**Fig 11. Restricted shear process diagram of rubber asphalt mortar.** (a) Rubber powder content of 0% (b) Rubber powder content of 10% (c) Rubber powder content of 20% (d) Rubber powder content of 30%.

where $F_f$ is friction force and $F_n$ is normal force.

$$F_f = F_0 + \tau A_c(L) \tag{6}$$

$$F_n = \phi A_c(L) \tag{7}$$

where $F_0$ and $\tau$ are Derjaguin offset and effective shear strength, respectively, which are independent of load, $A_c(L)$ is the contact area under load L, and $\varphi$ is the normal pressure.

The friction coefficients of rubber asphalt mortar with different rubber powder content and rubber asphalt mortar with 10% rubber powder content at -20~60°C are shown in Tables 6 and 7, respectively. It can be seen from Table 6 that the friction coefficient between the rubber asphalt mortar and the rubber interface is positively correlated with the amount of rubber powder. Similar to the effect of temperature on the Young's modulus of asphalt mortar, the Young's modulus of rubber asphalt mortar decreases with the increase of rubber powder content. The lower the Young's modulus, the more likely the rubber asphalt mortar is to deform during the friction process, increasing the effective contact area with the rubber surface, resulting in an increase in the friction coefficient.

It can be seen from Table 7 that when the content of rubber powder is 10%, the friction coefficient between rubber asphalt mortar and rubber also increases with the increase of temperature. At -20°C, the friction coefficient between rubber asphalt mortar and rubber is only 0.64. When the temperature rises to 60°C, the friction coefficient reaches nearly 1.45. This shows that the temperature has a great influence on the friction performance of rubber asphalt mortar and rubber.

**4.3.2. Transition mechanism of adhesive friction performance.** The simulated and measured values of the friction coefficient of rubber asphalt mortar are shown in Fig 12. It can be seen from the Fig that under the same temperature conditions, the simulated and measured values of rubber asphalt mortar increase with the increase of rubber powder content, and the increase gradually decreases. At the same time, at 0%, the measured value is greater than the simulated value. At 10%, the measured value is the same as the simulated value. At 20% and 30%, the measured values are smaller than the simulated values. This shows that the increase of the simulated value is greater than the measured value, which may be related to the increase of the number of rubber powder particles and the improvement of the surface roughness of the rubber asphalt mortar. At 30%, the difference between the simulated value and the measured value of the friction coefficient is the largest, and the maximum difference is 0.14. The raw data for Fig 12 are available in S5 Fig.

**Table 6. Friction coefficient of rubber asphalt mortar with different rubber powder content.**

| Rubber powder content | 0% | 10% | 20% | 30% |
|---|---|---|---|---|
| τ | 0.015 | 0.065 | 0.046 | 0.164 |
| φ | 0.095 | 0.14 | 0.089 | 0.202 |
| $A_c$ | 2237.385 | 2150.919 | 2562.617 | 2562.617 |
| Friction coefficient | 0.534274 | 0.729953 | 0.867619 | 0.966426 |

**Table 7. Friction coefficient of rubber asphalt mortar at different temperatures.**

| Temperature | -20°C | -10°C | 0°C | 10°C | 20°C | 30°C | 40°C | 50°C | 60°C |
|---|---|---|---|---|---|---|---|---|---|
| τ | 0.058 | 0.062 | 0.065 | 0.072 | 0.085 | 0.101 | 0.061 | 0.092 | 0.057 |
| φ | 0.162 | 0.154 | 0.14 | 0.139 | 0.153 | 0.144 | 0.09 | 0.105 | 0.065 |
| Ac | 2150.91 | 2150.91 | 2150.91 | 2150.91 | 2150.91 | 2150.91 | 2150.91 | 2150.91 | 2150.9 |
| Friction coefficient | 0.6425 | 0.64411 | 0.7299 | 0.7855 | 0.7986 | 1.0596 | 1.3910 | 1.4304 | 1.6491 |

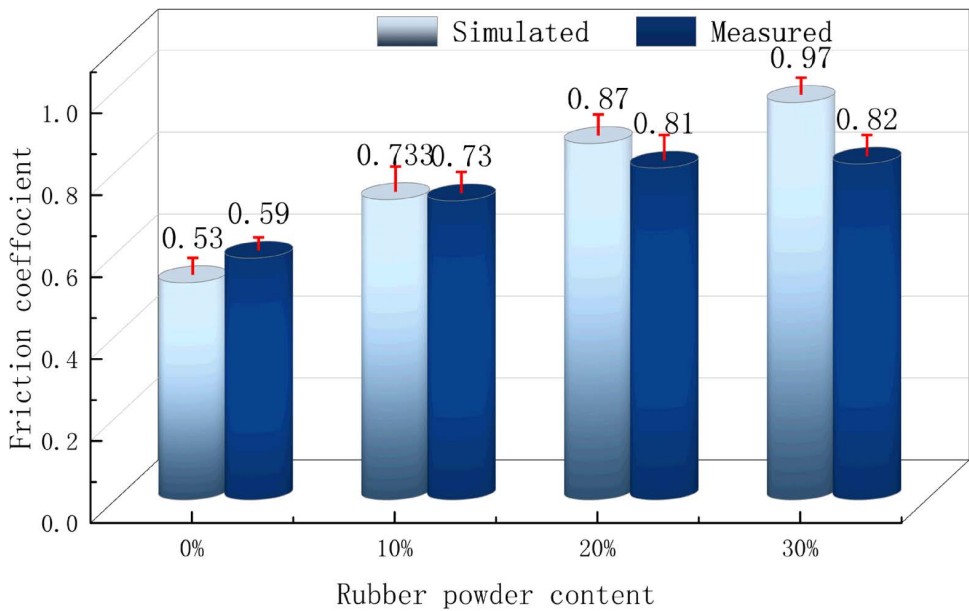

**Fig 12. Friction coefficient of asphalt mortar with different rubber content.**

Under different temperature conditions, the internal molecular chain configuration of asphalt and rubber is different, and the thermal motion state of internal molecules is different. From low temperature to high temperature, rubber asphalt mortar will show three different states: glassy state, high elastic state and viscous flow state [40]. Therefore, different ambient temperatures will have a greater impact on the mechanical properties of rubber. In order to reveal the influence of temperature on the friction performance of rubber asphalt mortar. Taking the rubber asphalt mortar model with rubber powder content of 10% and ambient temperature of -20°C, -10°C, 0°C, 10°C, 20°C, 30°C, 40°C, 50°C and 60°C as an example, the friction process is simulated to discuss the influence of ambient temperature on the friction performance of rubber asphalt mortar.

When the rubber powder content is constant, it can be observed from Fig 13 that as the temperature increases, the friction coefficient of the rubber asphalt mortar gradually increases. The friction coefficient is only 0.64 at -20°C. When the temperature rises to 60°C, the friction coefficient increases to 1.64, which is more than doubled at -20°C. At the same time, comparing the simulated value with the measured value, it is found that the simulated value is greater than the measured value at low temperature. With the increase of temperature, the growth rate of the measured value is greater, exceeding the simulated value at 20°C, and the difference between the simulated value and the measured value is the largest at 50°C, and the maximum difference is 0.32. The raw data for Fig 13 are available in S6 Fig.

The absolute value of the adhesion work between rubber asphalt mortar and rubber interface is shown in Fig 14. The main reason for the increase of friction coefficient is that with the increase of temperature, the rubber modified asphalt changes from glassy state to high elastic state and then to viscous flow state, and the adhesion between asphalt mortar and rubber interface is improved. The adhesion work can be used to indicate the adhesion performance of the interface between the two systems. The greater the absolute value, the better the adhesion of the two systems, and the less likely the interface of the system is to be destroyed [41]. The calculation equation is Eqs (8) and (9). The raw data for Fig 14 are available in S7 Fig.

$$W = \frac{E_a}{A} \tag{8}$$

$$E_a = E_{P+S} - (E_P + E_S) \tag{9}$$

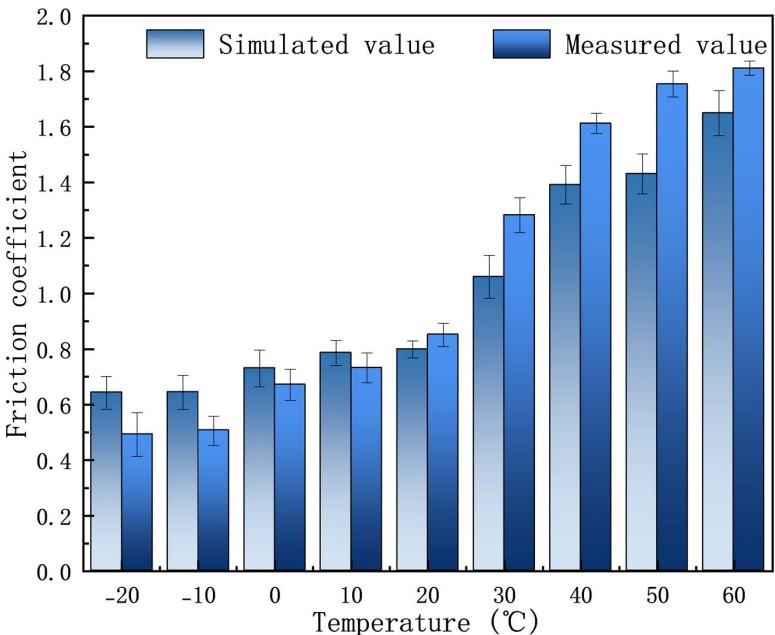

**Fig 13. Friction coefficient of asphalt mortar at different temperatures.**

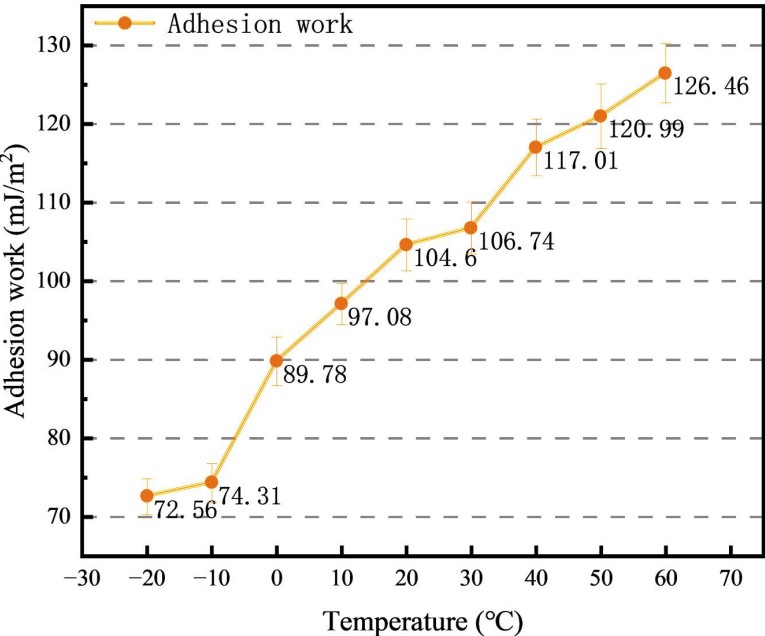

**Fig 14. Interfacial adhesion work between asphalt mortar and rubber at different temperatures.**

where W is the adhesion work between asphalt mortar and rubber interface; $E_a$ is the interaction energy of asphalt mortar-rubber interface after aggregate removal; $E_{P+S}$ is the total potential energy of asphalt mortar-rubber interface after aggregate removal; $E_p$ is the potential energy of asphalt mortar; $E_s$ is the potential energy of rubber.

When the ambient temperature is low, the rubber asphalt mortar is in the glassy state, the Young's modulus is large, and the anti-deformation ability is correspondingly strong. It will not be pushed and accumulated due to friction, and the movement of rubber on the surface of the rubber asphalt mortar is relatively smooth. With the increase of temperature, when the actual temperature exceeds the glass transfer temperature, the rubber asphalt mortar transforms into a high elastic state and has a certain fluidity. In the process of friction, some rubber asphalt mortar will adhere to the rubber surface, which hinders the translation of rubber, so the friction coefficient increases. When the temperature continues to rise, the rubber asphalt mortar changes into a viscous flow state, and it has been tightly adhered to before the rubber surface moves. The friction coefficient of rubber slider and rubber asphalt mortar is further increased because of the great adhesion force in the friction process.

## 5. Conclusions

In this study, Materials Studio 2020 software was used to construct the molecular model system of rubber asphalt mortar and rubber asphalt mortar-aggregate interface under the condition of real powder-binder ratio, and the model was verified and the physical properties were simulated. Secondly, on this basis, a three-layer friction pair model of rubber-asphalt mortar-aggregate is established. Finally, the adhesive friction performance of the interface friction system is simulated, and the variation law of the friction performance of rubber asphalt mortar under different temperature conditions and different rubber powder content conditions is analyzed, and the mechanism of the adhesive friction performance of rubber asphalt mortar is explained.

(1) The increase in temperature promotes phase transition and the enhancement of adhesive, which improves the friction performance of rubber asphalt mortar. As the temperature rises, the friction coefficient between rubber asphalt mortar and the rubber plane shows an increasing trend. The degree of fitting between the simulated values and the predicted values is good, with a maximum difference of 0.32.

(2) The addition of rubber powder helps improve the mechanical properties of the mortar. As the rubber powder content and temperature increase, both the bulk modulus and Young's modulus of rubber asphalt mortar show an upward trend.

(3) Based on the simulation and analysis of the three-layer friction molecular dynamics model of rubber-asphalt mortar-aggregate, it can be seen that the addition of rubber powder optimizes the friction performance of the mortar to a certain extent.

(4) Through the analysis of molecular order and density, it has been verified that the molecular models of rubber asphalt mortar constructed under different rubber powder contents (0%, 10%, 20% and 30%) can truly and reliably reflect the physical properties of rubber asphalt mortar.

## Supporting information

**S1 Fig. Raw data for** Fig 6.
(XLSX)

**S2 Fig. Raw data for** Fig 7.
(XLSX)

**S3 Fig. Raw data for** Fig 8.
(XLSX)

**S4 Fig. Raw data for** Fig 9.
(XLSX)

**S5 Fig. Raw data for** Fig 12**.**
(XLSX)

**S6 Fig. Raw data for** Fig 13**.**
(XLSX)

**S7 Fig. Raw data for** Fig 14**.**
(XLSX)

## Author contributions

**Conceptualization:** Yuan Du.

**Data curation:** Yuan Du, Ning Li, Xinyuan Liu.

**Methodology:** Yuan Du, Xinyuan Liu.

**Supervision:** She Shan.

**Validation:** Ning Li, Hui Dou.

**Visualization:** She Shan, Manbin Yang.

**Writing – original draft:** Yuan Du.

**Writing – review & editing:** Ning Li, Hui Dou, Yongzhi Chen, Manbin Yang.

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
