## [Decision Letter · Decision Letter 0]

10 Dec 2024

Dear Dr. Li,

Thank you for submitting your manuscript to PLOS ONE. After careful consideration, we feel that it has merit but does not fully meet PLOS ONE’s publication criteria as it currently stands. Therefore, we invite you to submit a revised version of the manuscript that addresses the points raised during the review process.

One reviewer suggested to reject your manuscript, because he considered your manuscript was more like a reprot. Please explain carefully.

We look forward to receiving your revised manuscript.

Kind regards,

Jiaolong Ren

Academic Editor

PLOS ONE

Journal Requirements: When submitting your revision, we need you to address these additional requirements. 1. Please ensure that your manuscript meets PLOS ONE's style requirements, including those for file naming. The PLOS ONE style templates can be found at https://journals.plos.org/plosone/s/file?id=wjVg/PLOSOne_formatting_sample_main_body.pdf and https://journals.plos.org/plosone/s/file?id=ba62/PLOSOne_formatting_sample_title_authors_affiliations.pdf 2. Thank you for stating the following financial disclosure: "The National Natural Science Foundation of China (51868042), Special Funds for Guiding Local Scientific and Technological Development by The Central Government (22ZY1QA005), Gansu Science and Technology Major Project (22ZD6GA010), Gansu Provincial Key R&D Program(22YF7GA135)�the Key Research and Development Program of Ningxia Autonomous Region of China (2022BDE02010), and Lanzhou Science and Technology Project (2022-2-72). Gansu province science and technology plan project(21YF5GA001)."  Please state what role the funders took in the study.  If the funders had no role, please state: ""The funders had no role in study design, data collection and analysis, decision to publish, or preparation of the manuscript."" If this statement is not correct you must amend it as needed. Please include this amended Role of Funder statement in your cover letter; we will change the online submission form on your behalf. 3. Thank you for stating the following in the Acknowledgments Section of your manuscript: "The authors gratefully acknowledge the financial supports by the National Natural Science Foundation of China (51868042), Special Funds for Guiding Local Scientific and Technological Development by The Central Government (22ZY1QA005), Gansu Science and Technology Major Project (22ZD6GA010), Gansu Provincial Key R&D Program(22YF7GA135)�the Key Research and Development Program of Ningxia Autonomous Region of China (2022BDE02010), and Lanzhou Science and Technology Project (2022-2-72). Gansu province science and technology plan project(21YF5GA001)." We note that you have provided funding information that is not currently declared in your Funding Statement. However, funding information should not appear in the Acknowledgments section or other areas of your manuscript. We will only publish funding information present in the Funding Statement section of the online submission form. Please remove any funding-related text from the manuscript and let us know how you would like to update your Funding Statement. Currently, your Funding Statement reads as follows: "The National Natural Science Foundation of China (51868042), Special Funds for Guiding Local Scientific and Technological Development by The Central Government (22ZY1QA005), Gansu Science and Technology Major Project (22ZD6GA010), Gansu Provincial Key R&D Program(22YF7GA135)�the Key Research and Development Program of Ningxia Autonomous Region of China (2022BDE02010), and Lanzhou Science and Technology Project (2022-2-72). Gansu province science and technology plan project(21YF5GA001)." Please include your amended statements within your cover letter; we will change the online submission form on your behalf. 4. We note that your Data Availability Statement is currently as follows: All relevant data are within the manuscript and its Supporting Information files. Please confirm at this time whether or not your submission contains all raw data required to replicate the results of your study. Authors must share the “minimal data set” for their submission. PLOS defines the minimal data set to consist of the data required to replicate all study findings reported in the article, as well as related metadata and methods (https://journals.plos.org/plosone/s/data-availability#loc-minimal-data-set-definition). For example, authors should submit the following data: - The values behind the means, standard deviations and other measures reported;- The values used to build graphs;- The points extracted from images for analysis. Authors do not need to submit their entire data set if only a portion of the data was used in the reported study. If your submission does not contain these data, please either upload them as Supporting Information files or deposit them to a stable, public repository and provide us with the relevant URLs, DOIs, or accession numbers. For a list of recommended repositories, please see https://journals.plos.org/plosone/s/recommended-repositories. If there are ethical or legal restrictions on sharing a de-identified data set, please explain them in detail (e.g., data contain potentially sensitive information, data are owned by a third-party organization, etc.) and who has imposed them (e.g., an ethics committee). Please also provide contact information for a data access committee, ethics committee, or other institutional body to which data requests may be sent. If data are owned by a third party, please indicate how others may request data access. 5. PLOS requires an ORCID iD for the corresponding author in Editorial Manager on papers submitted after December 6th, 2016. Please ensure that you have an ORCID iD and that it is validated in Editorial Manager. To do this, go to ‘Update my Information’ (in the upper left-hand corner of the main menu), and click on the Fetch/Validate link next to the ORCID field. This will take you to the ORCID site and allow you to create a new iD or authenticate a pre-existing iD in Editorial Manager.

Reviewers' comments:

**Comments to the Author**

1. Is the manuscript technically sound, and do the data support the conclusions?

Reviewer #1: Partly

Reviewer #2: Yes

2. Has the statistical analysis been performed appropriately and rigorously?

Reviewer #1: No

Reviewer #2: Yes

3. Have the authors made all data underlying the findings in their manuscript fully available?

Reviewer #1: No

Reviewer #2: Yes

4. Is the manuscript presented in an intelligible fashion and written in standard English?

Reviewer #1: Yes

Reviewer #2: Yes

Reviewer #1: Dear

The manuscript presents a good and important qualitative work that attempts to fill the knowledge gap in the partial composition of the asphalt mixture components. It did not present a new proposal and was satisfied with listing the mechanism for applying or using the program. Materials Stoduo

The manuscript is written in a format closer to a report or guide to represent the partial components of the asphalt mixture.

It is very important that the real practical importance of the aim of the current study

Reviewer #2: Title: Change the title to be ((temperature dependence of adhesive friction properties of rubber asphalt mortar and its mechanism of action))

Abstract: Remove the first sentence ((Abstract: Excellent pavement skid resistance guarantees the friction resistance required for braking between tires and roads.))

Conclusions rather than ((conclusion)). The conclusions must expose what was found from behind the results; not repeating the results only. Rewrite the conclusions.

**Do you want your identity to be public for this peer review?** For information about this choice, including consent withdrawal, please see our Privacy Policy

Reviewer #1: **Yes: ** ABDULHAQ HADI ABED ALI AL-HADDAD

Reviewer #2: **Yes: ** Prof. Dr. Ahmed Mancy Mosa

---

## [Author Response · Author response to Decision Letter 1]

15 Feb 2025

Response to Reviewers

Dear Editors and Reviewers:

Thank you for your letter and for the reviewers’ comments concerning our manuscript entitled ' Study on the temperature dependence of adhesive friction properties of rubber asphalt mortar and its mechanism of action' (Manuscript Number: PONE-D-24-53366). Those comments are all valuable and helpful for revising and improving our paper, as well as an important guide for our research. We have studied the comments carefully and made revisions. Revised parts are marked in red on the paper. Our responses to reviewers' comments are summarised below.

I would like to thank the editors and reviewers for their work and thank you for giving us the opportunity to be able to submit a revised version of this paper. Thank you again for your comments and suggestions.

Sincerely yours,

Ning Li

Response to Reviewers #1:

Point 1: The manuscript presents a good and important qualitative work that attempts to fill the knowledge gap in the partial composition of the asphalt mixture components. It did not present a new proposal and was satisfied with listing the mechanism for applying or using the program.

Response 1: Thank you very much for your valuable comments, which have significantly improved the quality of our paper. Regarding your point that the manuscript did not present a new proposal and was only satisfied with listing the mechanism for applying or using the program, I fully understand your concerns. However, the main focus of this paper is to establish a three - layer friction pair model of rubber - asphalt mortar - aggregate, simulate the adhesive friction performance of the interface friction system, analyze the variation laws of the friction performance of rubber asphalt mortar under different temperature conditions and different rubber powder content conditions, and explain the mechanism of the adhesive friction performance of rubber asphalt mortar. Based on the new proposal you mentioned about the temperature - dependence of the adhesive friction performance of rubber asphalt mortar mixtures and its mechanism, it remains an important issue that requires further exploration. In light of your valuable suggestions, we have reviewed a large number of literatures. In the subsequent research, the authors will conduct in - depth analysis and discussion on the new proposal regarding the temperature - dependence of the adhesive friction performance of rubber asphalt mortar and its mechanism. We sincerely appreciate your guidance and will continue to strive to improve the quality of our research.

Once again, the authors would like to thank you for your valuable comments and for reviewing the manuscript!

Point 2: The manuscript is written in a format closer to a report or guide to represent the partial components of the asphalt mixture.

Response 2: Thank you for your valuable comments, which have significantly enhanced the quality of our paper. Regarding your concern that the writing format of this manuscript is more akin to a report or guide introducing the partial components of the asphalt mixture, I fully understand. In this study, however, the authors utilized Materials Studio 2020 software to construct the molecular model systems of rubber asphalt mortar and the rubber asphalt mortar - aggregate interface under the actual powder - binder ratio conditions. The models were verified, and their physical properties were simulated. Subsequently, on this basis, a three - layer friction pair model of rubber - asphalt mortar - aggregate was established. Finally, the adhesive friction performance of the interface friction system was simulated, the variation laws of the friction performance of rubber asphalt mortar under different temperature conditions and different rubber powder content conditions were analyzed, and the mechanism of the adhesive friction performance of rubber asphalt mortar was explained. We sincerely appreciate your guidance and will continue to strive to improve the quality of our research.

Once again, the authors would like to thank you for your valuable comments and for reviewing the manuscript!

Point 3: It is very important that the real practical importance of the aim of the current study.

Response 3: Thank you very much for your meticulous guidance on this manuscript. We have carefully considered and analyzed the issues you raised regarding this manuscript. This study is of great significance in practical applications, which is mainly reflected in the following aspects: The adhesive friction performance of rubber asphalt mortar directly affects the skid resistance and durability of roads. By simulating the variation laws of friction performance under different conditions, the research can provide more adaptable materials for road construction in different climate regions, ensuring that roads can maintain good performance under various temperatures. We sincerely appreciate your guidance and will continue to strive to improve the quality of our research.

Once again, the authors would like to thank you for your valuable comments and for reviewing the manuscript! 

Response to Reviewers #2:

Point 1: Title: Change the title to be ((temperature dependence of adhesive friction properties of rubber asphalt mortar and its mechanism of action).

Response 1: Thank you very much for your careful guidance on this manuscript. We have carefully considered and analyzed the issues you raised about this manuscript. We agree with your opinion. We have carefully and specifically elaborated in the text and replaced the relevant sections with red markings. Your comments have greatly improved the quality of our manuscript!

Point 2: Abstract: Remove the first sentence ((Abstract: Excellent pavement skid resistance guarantees the friction resistance required for braking between tires and roads.))

Response 2: Thank you very much for your meticulous guidance on this manuscript. We have carefully considered and analyzed the issues you raised regarding this manuscript. We have deleted the first sentence of the abstract ("Excellent pavement skid resistance guarantees the friction resistance required for braking between tires and roads"). Your comments have significantly enhanced the quality of our manuscript!

Point 3: Conclusions rather than ((conclusion)). The conclusions must expose what was found from behind the results; not repeating the results only. Rewrite the conclusions.

Response 3: Thank you for your valuable comments, which have significantly enhanced the quality of the paper. In response to your concerns, the authors have carefully re-summarized the conclusions to expose the discoveries behind the research results, rather than merely repeating the results. The revised parts in the paper are marked in red. The specific revised content is as follows:

Revised version:

(1) The increase in temperature promotes phase transition and the enhancement of adhesive, which improves the friction performance of rubber asphalt mortar. As the temperature rises, the friction coefficient between rubber asphalt mortar and the rubber plane shows an increasing trend. The degree of fitting between the simulated values and the predicted values is good, with a maximum difference of 0.32.

(2) The addition of rubber powder helps improve the mechanical properties of the mortar. As the rubber powder content and temperature increase, both the bulk modulus and Young's modulus of rubber asphalt mortar show an upward trend.

(3) Based on the simulation and analysis of the three-layer friction molecular dynamics model of rubber-asphalt mortar-aggregate, it can be seen that the addition of rubber powder optimizes the friction performance of the mortar to a certain extent.

(4) Through the analysis of molecular order and density, it has been verified that the molecular models of rubber asphalt mortar constructed under different rubber powder contents (0%, 10%, 20% and 30%) can truly and reliably reflect the physical properties of rubber asphalt mortar.

---

## [Decision Letter · Decision Letter 1]

3 Mar 2025

Temperature dependence of adhesive friction properties of rubber asphalt mortar and its mechanism of action

PONE-D-24-53366R1

Dear Dr. Li,

We’re pleased to inform you that your manuscript has been judged scientifically suitable for publication and will be formally accepted for publication once it meets all outstanding technical requirements.

Kind regards,

Jiaolong Ren

Academic Editor

PLOS ONE

Additional Editor Comments (optional):

Reviewers' comments:

Reviewer's Responses to Questions

**Comments to the Author**

Reviewer #1: (No Response)

Reviewer #2: All comments have been addressed

2. Is the manuscript technically sound, and do the data support the conclusions?

Reviewer #1: Yes

Reviewer #2: Yes

3. Has the statistical analysis been performed appropriately and rigorously?

Reviewer #1: Yes

Reviewer #2: Yes

4. Have the authors made all data underlying the findings in their manuscript fully available?

Reviewer #1: Yes

Reviewer #2: Yes

5. Is the manuscript presented in an intelligible fashion and written in standard English?

Reviewer #1: Yes

Reviewer #2: Yes

Reviewer #1: Thank you for your corrections and Good luck

Abdulhaq

Reviewer #2: The authors have adequately addressed the comments stated in the first review. The manuscript is now acceptable for publication.

After the response of the authors, the manuscript describes a technically sound piece of scientific research with data that supports the conclusions. Experiments have been conducted rigorously, with appropriate controls, replication, and sample sizes. The conclusions was drawn appropriately based on the data presented. Now and after corrections, the authors made all data underlying the findings described in the manuscript fully available without restriction. Overall, I am satisfied with the manuscript in its present form. Therefore, the manuscript can be published in PLOS one.

**Do you want your identity to be public for this peer review?** For information about this choice, including consent withdrawal, please see our Privacy Policy

Reviewer #1: **Yes: ** Prof dr.abdulhaq Hadi abedali

Reviewer #2: **Yes: ** Prof. Dr. Ahmed mancy Mosa

---

## [Editor Report · Acceptance letter]

PONE-D-24-53366R1

PLOS ONE

Dear Dr. Li,

I'm pleased to inform you that your manuscript has been deemed suitable for publication in PLOS ONE. Congratulations! Your manuscript is now being handed over to our production team.

Kind regards,

on behalf of

Dr. Jiaolong Ren

Academic Editor

PLOS ONE